# Internalization of Neutrophil-Derived Microvesicles Modulates TNFα-Stimulated Proinflammatory Cytokine Production in Human Fibroblast-Like Synoviocytes

**DOI:** 10.3390/ijms22147409

**Published:** 2021-07-10

**Authors:** Dong Zhan, Andrew Cross, Helen L. Wright, Robert J. Moots, Steven W. Edwards, Sittisak Honsawek

**Affiliations:** 1Joint PhD Program in Biomedical Sciences and Biotechnology, Faculty of Medicine, Chulalongkorn University, Bangkok 10330, Thailand; Dong.Zhan@liverpool.ac.uk; 2Institute of Ageing and Chronic Disease, University of Liverpool, Liverpool L7 8TX, UK; across@liverpool.ac.uk (A.C.); hlwright@liverpool.ac.uk (H.L.W.); R.J.Moots@liverpool.ac.uk (R.J.M.); 3Human Anatomy Laboratory of Experimental Teaching Centre, School of Basic Medical Sciences, Kunming Medical University, Kunming 650500, China; 4Department of Academic Rheumatology, Aintree University Hospital, Liverpool L9 7AL, UK; 5Institute of Integrative Biology, University of Liverpool, Liverpool L69 7ZB, UK; S.W.Edwards@liverpool.ac.uk; 6Osteoarthritis and Musculoskeleton Research Unit, Department of Biochemistry, Faculty of Medicine, Chulalongkorn University, Bangkok 10330, Thailand; 7Vinai Parkpian Orthopaedic Research Centre, Department of Orthopaedics, King Chulalongkorn Memorial Hospital, Chulalongkorn University, Bangkok 10330, Thailand

**Keywords:** fibroblast-like synoviocytes, neutrophil-derived microvesicles, cytokines, synovitis, osteoarthritis

## Abstract

Neutrophil-derived microvesicles (NDMVs) have the potential to exert anti-inflammatory effects. Our study aimed to explore the effects of NDMVs on proinflammatory cytokines expressed by tumor necrosis factor α (TNFα)-stimulated fibroblast-like synoviocytes (FLS). FLS were isolated from the synovium of knee osteoarthritis (OA) patients undergoing surgery. NDMVs, isolated from TNFα-stimulated healthy neutrophils, were characterized by electron microscopy and nanoparticle tracking analysis. MTT and scratch wound healing assays were used to measure FLS viability and migration after treatment with NDMVs, while internalization of fluorescently labeled NDMVs was appraised by flow cytometry and confocal microscopy. Levels of proinflammatory cytokines in supernatants were quantified by the Bio-Plex system. Incubation of FLS with NDMVs at a vesicle/cell ratio of 100 resulted in a time-dependent uptake, with 35% of synoviocytes containing microvesicles over a 6–24 h time period, with no significant change in cell viability. TNFα stimulated the cytokine expression in FLS, and NDMVs down-regulated TNFα-induced expression of IL-5, IL-6, IL-8, MCP-1, IFNγ and MIP-1β. However, this down-regulation was selective, as NDMVs had no significant effects on TNFα-stimulated expression of IL-2 or IL-4. NDMVs were internalized by FLS to inhibit TNFα-stimulated broad-spectrum proinflammatory cytokine secretion. NDMVs, therefore, may exhibit an anti-inflammatory role in the regulation of the FLS function.

## 1. Introduction

Osteoarthritis (OA) is a degenerative disease of synovial joints resulting from cartilage loss that manifests as articular pain and osteophytes. As a weight-bearing and structurally complex joint, the knee joint is at higher risk of developing OA, which was originally considered to stem from the “wear and tear” of articular cartilage but is now characterized by low-grade and chronic inflammation, especially in the synovium [1,2]. OA synovitis, therefore, contributes to the pathological and pathophysiological progress of this joint disease. Fibroblast-like synoviocytes (FLS) are the main cellular type in the intima layer of the synovium and they regulate homeostasis and tissue repair within the synovial joint. FLS secrete hyaluronan and proteoglycan-4 to lubricate joints but also produce a variety of inflammatory and angiogenic cytokines/chemokines and matrix metalloproteinases (MMPs) that regulate articular inflammation and joint function [3,4,5]. 

Extracellular vesicles (EVs) are released from a variety of cells and are membrane-bound structures containing a variety of cellular components that have the potential to modify the properties of target cells that absorb them. There are three major types of EVs: exosomes, microvesicles (MVs) and apoptotic bodies, which are characterized by their size and composition. Microvesicles are between 100 and 1000 nm in diameter, which are larger than exosomes (typically 40–120 nm) but smaller than apoptotic bodies (500–2000 nm) [6]. Donor cells generate MVs via a budding method prior to their release into the extracellular space and have been detected in the synovial fluid of patients with rheumatoid arthritis (RA), OA and juvenile idiopathic arthritis [7]. Neutrophil-derived microvesicles (NDMVs) can accumulate at inflammatory or infectious sites at higher levels than other leukocyte-derived MVs [8], and monocytes and dendritic cells have been shown to be modulated by NDMVs to up-regulate the expression of transforming growth factor β (TGFβ) and down-regulate the expression of proinflammatory cytokines including interleukin-6 (IL-6), IL-8 and tumor necrosis factor α (TNFα) [6]. Annexin A1 expressed on NDMVs can destabilize the adhesion of leukocytes to endothelial cells and prevent inflammation [9]. NDMVs have also been delineated to block the polarization of monocyte-derived macrophages to indirectly influence the activation of FLS [10]. In addition to their anti-inflammatory properties, NDMVs can regulate chondrocyte function by decreasing IL-8 and prostaglandin E2 release, and by increasing extracellular matrix components to maintain cartilage function [11].

In view of the potential of NDMVs to exert anti-inflammatory effects, our study aimed to determine if these structures are internalized by FLS isolated from OA patients and to explore whether this uptake could regulate the ability of the cells to generate cytokines and chemokines in response to TNFα. The present study unveiled that FLS internalized NDMVs to selectively down-regulate the expression of a range of TNFα-induced cytokines and therefore may mediate anti-inflammatory effects under these conditions that may limit local synovitis.

## 2. Results

### 2.1. Morphological and Size Characteristics of NDMVs

Transmission electron microscopy (TEM) evidenced that the isolated NDMVs had a cup-like structure (minus a handle), and their size ranged from 200 to 500 nm (Figure 1A). Scanning electron microscopy (SEM) revealed that the NDMVs had an irregular spherical appearance, with an uneven surface, again with their size range largely between 200 and 500 nm (Figure 1B), but some vesicles were smaller than 100 nm. Nanoparticle tracking analysis (NTA), which measures the Brownian motion of NDMVs (Figure 1C), illustrated that the diameter ranged from 38.4 to 1000.0 nm, with a mean value of 232.3 ± 8.9 nm. These measurements provided better quantification of the numbers and sizes of the NDMVs and evinced that those particles with a diameter of <100 nm comprised approximately 2–3% of the total population. The concentration of NDMVs was calculated as 3.8 × 10^8^ (±1.9 × 10^7^) particles/mL by this method. This analysis disclosed that NDMVs were identifiable as four discrete sub-populations, based on their size (Figure 1D), with the greatest proportion having a diameter between 100 and 150 nm.

### 2.2. Effect of NDMVs on FLS Viability

The viability of FLS treated with 10 ng/mL TNFα for 24 h was not significantly different to that of untreated controls in the absence of TNFα (Figure 2A). However, when FLS were exposed to NDMVs at vesicle/cell (V/C) ratios of 200, 300 and 400 for 24 h, the cell viabilities were decreased to 63%, 58% and 69%, respectively (*p* < 0.01). However, after incubation of FLS with NDMVs at ratios of 10, 30, 50 and 100 per FLS for 24 h, there were no significant differences in the viabilities compared to controls (Figure 2A). FLS were therefore incubated with NDMVs at a V/C ratio of 100, and viability was measured over a 48 h period. We observed that FLS viability was not significantly affected by incubation with this concentration of NDMVs (V/C ratio of 100) for up to 24 h of incubation, but by 48 h of incubation, FLS viability was significantly decreased (*p* = 0.008, Figure 2B). Therefore, in subsequent experiments, FLS were incubated with NDMVs at a V/C ratio of 100 for 24 h, as these experimental conditions had no measurable impact on FLS viability.

### 2.3. Effect of NDMVs on FLS Apoptosis

Subsequently, we inspected whether NDMVs down-regulate FLS viability and induce FLS apoptosis by a terminal deoxynucleotidyl transferase dUTP nick-end labeling (TUNEL) assay and caspase-3 activity analysis. Figure 3A portrays representative images of TUNEL staining in each group, showing that apoptotic cell nuclei were stained brown or dark brown. We found that TUNEL-positive cells were obviously increased in FLS treated with NDMVs, especially at a V/C ratio of 300 for 24 h, compared to the control group (*p* < 0.001) (Figure 3B). After NDMV co-culture treatment extended to 48 h, the TUNEL-positive cells in the NDMV treatment with V/C ratios of 100 and 300 were substantially greater than those in the control group (*p* < 0.001) (Figure 3A,B). 

Further, the caspase-3 activity assay, consistent with the TUNEL findings, revealed that the relative caspase-3 activity of FLS treated with NDMVs at a V/C ratio of 300 for 24 h was significantly higher than that of the 24 h control group (*p* < 0.05) (Figure 3C). Moreover, the relative caspase-3 activity of FLS in NDMV treatment at a V/C ratio of 300 was significantly greater than that of the 48 h control group, as delineated in Figure 3C (*p* < 0.05).

### 2.4. Effect of NDMVs on FLS Migration

The wound healing area (WHA) of FLS with/without NDMV treatment is displayed in Figure 4A. When exposed to NDMVs at a V/C ratio of 100 for 24 h, although the WHA of FLS was reduced to 16.0 ± 4.0%, compared to 27.0 ± 3.1% WHA of FLS without exposure to NDMVs, there was no significant difference (*p* = 0.15). At 48 h, the WHA of FLS exposed to NDMVs was 13.0 ± 4.5%, which was significantly lower than that of FLS without NDMV exposure (*p* = 0.008) (Figure 4B). Meanwhile, FLS exposed to NDMVs for 48 h were decreased in the cellular population, and their nuclei were stained darker (Figure 4A).

### 2.5. FLS Internalize NDMVs

FLS were treated with PKH26-labeled NDMVs at a V/C ratio of 100 for different time periods, and internalization was analyzed initially by flow cytometry. Uptake of labeled NDMVs by FLS was detected as early as 0.5 h after addition and was enhanced by 6 h of incubation, but uptake was not increased further by 24 h of incubation (Figure 5A). After 1 h of incubation, 3.6% of FLS took up NDMVs, and by 6 h of incubation, over 35% of the cells stained positive for NDMV uptake, while by 24 h of incubation, 37% of the FLS stained positive. Calculation of uptake as a fold change in the mean fluorescence intensity (MFI) revealed a 1.25-fold increase by 0.5 h (compared to controls), rising to a 2.4-fold increase by 6 h, which was largely unaltered by 24 h (2.1-fold) and 48 h (2.6-fold), as illustrated in Figure 5B.

Confocal microscopy analysis substantiated that the detected fluorescent signal was indeed from the uptake of PKH26-labeled NDMVs by FLS, and not from NDMVs bound to the cell surface. Z-stack images were collected to reconstruct three-dimensional images and unveiled that single and multiple NDMVs were localized in the FLS cytoplasm in equivalent focal planes to the actin cytoskeleton (Figure 6A,B). Vertical sections generated by three-dimensional analysis of confocal sections affirmed that the majority of the fluorescent NDMV signal was indeed observed inside the cells. This experimental approach asserted that NDMVs were internalized by FLS in a time-dependent manner (Figure 6C).

### 2.6. Proinflammatory Cytokine Levels in FLS Culture Medium

We then assessed the effects of NDMVs on the ability of FLS to secrete a range of cytokines after incubation for 6 h and 24 h in the presence and absence of TNFα. In these experiments, we evaluated the effects of NDMVs alone on FLS cytokine expression and the ability of NDMVs to affect TNFα-regulated expression, after co-incubation of FLS with NDMVs plus TNFα. Of the cytokines analyzed, all except IL-1β, IL-10, IL-7, IL-12, IL-13 and GM-CSF were detected in FLS culture medium.

The addition of TNFα for 24 h alone significantly stimulated the secretion of IL-2, IL-4, IL-5, IL-6, IL-8, IFNγ, MCP-1 and MIP-1β (*p* < 0.05, Figure 7). TNFα also significantly elevated IL-17 and G-CSF expression by FLS, and while this increased expression was statistically significant, the magnitude of this increment was not as great as that for other cytokines and chemokines, mainly because of the more variable expression rates by unstimulated FLS. 

The supplement of NDMVs only to FLS for 6 h and 24 h had little effect on cytokine expression, except for a significant decline in endogenous IL-6, IFNγ and MCP-1 expression. However, when co-incubated with TNFα, NDMVs significantly subdued the TNFα-induced expression of IL-5, IL-6, IL-8, IL-17, G-CSF, IFNγ, MCP-1 and MIP-1β (*p* < 0.05, Figure 7). In contrast, NDMVs had no significant impact on TNFα-induced IL-2 and IL-4 expression, denoting their inhibitory effect on a broad spectrum of cytokines’ expression.

### 2.7. Heatmap and Cluster Analysis of Proinflammatory Cytokine Levels in FLS Culture Medium

The presentation of the levels of the 10 proinflammatory cytokines in the form of a color-coded heat map which was generated using hierarchical clustering exhibited a good overview of the profile differences between the groups, as depicted in Figure 8. Hierarchical cluster analysis via heatmaps evinced relationships between patterns of cytokine expression in the presence and absence of TNFα and NDMVs. This attests that cytokine expression in the absence of TNFα was primarily unaffected by treatment with NDMVs: the control and NDMV exposure for 6 h or 24 h were lower and these clustered together (Figure 8). However, in the presence of TNFα and NDMVs, the patterns of cytokine expression clustered, but two patterns were apparent: there was a down-regulated expression of IL-17, G-CSF, IL-8, MIP-1β, MCP-1, IL-5, IL-6 and IFNγ after NDMV exposure for 6 h and 24 h, while in contrast, IL-2 and IL-4 were down-regulated by NDMVs after 24 h treatment. 

## 3. Discussion

In this study, we have uncovered that NDMVs that are taken up by fibroblast-like synoviocytes from patients with OA selectively inhibit TNFα-dependent expression of a number of chemokines and cytokines. As TNFα drives inflammation in a number of inflammatory diseases, and OA is often associated with synovitis, these results corroborate the idea that NDMVs could have anti-inflammatory activity under these conditions. Our novel data add to the growing body of evidence to proclaim that, under certain experimental and perhaps disease conditions, NDMVs exert anti-inflammatory effects. They may thus have potential as novel adjuncts to therapeutic strategies in the treatment of OA. 

Classification of EVs is based mostly on their size, but there is considerable overlap in the sizes of different types of EVs, and these sizes may hinge on the method of isolation and purification, and the cell type producing them [12,13,14]. The conventional method employed to isolate EVs is differential centrifugation, and it has been reported that NDMV-enriched fractions may be prepared by centrifugation at 20,000 or 15,700× *g* [10,15]. However, NDMVs are more commonly provided by ultra-centrifugation at the conditions that we used in this study: 100,000× *g* [11,16,17,18]. The NTA results unveil that 97–98% of NDMVs that we prepared were in the size range of 100–1000 nm, which was affirmed by TEM and SEM. Therefore, the morphology and size characteristics of NDMVs used in our experiments were consistent with our expectations and previous reports. 

NDMVs have been documented to express annexin A1, CD66b and CD62L on their surface (as do intact neutrophils) and also myeloperoxidase (MPO) [11,18] which may elicit oxidative damage and produce cytotoxic hypochlorous acid and other oxidants in inflammatory conditions [19]. MPO has been identified in the synovial fluid of OA and RA patients, and intra-articular injection of MPO has been documented to increase arthritis severity in mouse models [20,21]. Internalized NDMVs containing MPO could therefore induce MPO accumulation, and its intracellular activity might interfere with FLS viability, proliferation and migration. The scratch wound assay illustrated that NDMVs were capable of slowing down FLS migration, indicated by the percentage of the wound healing area in the 48 h NDMV treatment. Such a mechanism may, at least in part, explain why high-dose (V/C > 100) and long-term (>24 h) NDMV treatments adversely affected FLS viability. Our studies on the effect of NDMVs on the viability of FLS showed that a greater number of microvesicles predominantly affected the proliferation of the FLS. This process depended on the number of microvesicles added per FLS cell, with a higher V/C ratio than 100 resulting in a lower cell viability. The explanation for this finding could be that NDMVs deliver a range of inflammatory proteins, mRNA and numerous microRNAs and are internalized by FLS which could influence cell viability in recipient FLS. Additionally, NDMVs may modulate the proliferation of FLS by the generation of reactive oxygen intermediates including the release of MPO and hydrogen peroxide. It should be noted that NDMVs contain a complex mixture of proteins, RNAs, microRNAs and unknown gene materials that may interfere with the deregulated FLS survival. Therefore, future studies should address the potential role of these factors. Moreover, the MTT assay for cellular viability is contingent upon the reductive nature of nicotinamide adenine dinucleotide phosphate (NADPH) and the oxidative nature of MTT [22]. Exogenous MPO might promote the oxidative action of FLS to create more advanced oxidative protein products (AOPPs) to motivate NADPH oxidase, meaning that the NADPH concentration would be decreased [23]. This might result in a reduced level of purple formazan after treatment with a vesicle/cell ratio of 100 for 24 h. However, MPO from NDMVs may not influence FLS viability as no significant difference was observed at 24 h of treatment with NDMVs. After long-term exposure to NDMVs for 48 h, the nuclei of FLS were stained darker by crystal violet. Additionally, decreased cellular confluency and a reduced cell size were evident. By using light microscopy, we found that plenty of FLS floated in the culture medium. Therefore, long-term (48 h) treatment with NDMVs is harmful to FLS. 

To determine whether NDMVs down-regulate FLS viability and induce FLS apoptosis, we performed a TUNEL assay and caspase-3 activity analysis. TUNEL staining showed that the 24 and 48 h NDMV treatments with a V/C ratio of 300 promoted the apoptosis of FLS. In addition, activation of caspase-3 may result in apoptotic cell death. Subsequent analysis also demonstrated that caspase-3 activity was raised in FLS treated with NDMVs at a V/C ratio of 300 for 24 and 48 h, suggesting the induction of FLS apoptosis. High-ratio and long-term treatment of NDMVs could induce FLS apoptosis rather than down-regulation of the metabolic state of the cells.

Further analysis divulged that approximately 37% of FLS ingested labeled NDMVs over a 6 to 48 h time period. EV internalization is dependent on the EV dose, treatment time and the nature of the recipient cells. Several different mechanisms of endocytosis can be involved in the uptake of nanoparticles, including micropinocytosis, clathrin-dependent endocytosis and caveolin-dependent endocytosis [24]. A previous study outlined that up-regulated caveolin-dependent endocytosis significantly enhanced the uptake of dexamethasone-carbon nanotubes in TNFα-activated FLS, which in turn inhibited proinflammatory cytokines and MMPs [25]. Recently, Rattner et al. demonstrated that human synovium FLS, cultured human FLS and FLS cells expressed non-motile cilia that could play an endocytic function through the cilium pit [26]. It is proposed that primary cilia and the cilium pit in FLS function to regulate endocytosis including NDMV internalization. Additional research is required to determine the uptake mechanism of neutrophil-derived microvesicles that underpins their entry into FLS.

The proinflammatory cytokine TNFα is generated by a variety of cells including neutrophils, macrophages, lymphocytes and adipocytes to induce systemic inflammation [27]. TNFα has been detected in the synovial fluid, plasma and synovial membrane of OA patients and can therefore potentially activate both tissue cells and infiltrating immune cells [28,29,30,31,32]. In our experiments, NDMVs down-regulated the expression of a number of inflammatory cytokines and chemokines expressed by TNFα-activated FLS from OA patients. IL-7, IL-12 and IL-13 expression was undetectable in any experimental condition tested, and IL-5, IL-6, IFN-γ and MCP-1 expression levels were decreased by NDMVs in non-TNFα-activated FLS [33,34]. This finding was contradictory to that of Mesri et al., who reported that IL-6 and MCP-1 were elevated by NDMVs via the JNK1 signaling pathway in endothelial cells [35]. Our experiments suggest that NDMVs’ effects are attributed to the nature and function of the target cells, or the dose and time of treatments. TNFα-activated IL-8 expression was suppressed by NDMVs in line with previous research which described that NDMVs could decrease IL-8 secretion by macrophages activated with zymosan or lipopolysaccharide [36]. 

The present study unfolded that while NDMVs alone did not decrease IL-10 secretion, the TNFα-down-regulated expression of this anti-inflammatory cytokine from FLS was blocked by NDMVs. This phenomenon represents an additional regulatory role of NDMVs, by restoring the expression of an anti-inflammatory cytokine that was down-regulated in response to TNFα. Levels of the proinflammatory cytokine IL-17 are elevated in the serum, synovial fluid and inflamed synovium in OA and are associated with disease severity [37,38]. IL-17 can enhance MMP expression and up-regulate the expression of the proinflammatory cytokines IL-1, IL-6, IL-8 and TNFα in FLS from OA patients [39,40]. Interestingly, NDMVs up-regulated IL-17 levels released from FLS, but this effect was depressed in the presence of TNFα, which increased IL-17 expression after 24 h of incubation. The anti-inflammatory cytokine IL-4 was not altered in FLS incubated with NDMVs in the absence of TNFα, similar to the proinflammatory cytokine IL-2. However, IL-4 expression was decreased in FLS incubated with NDMVs in the presence of TNFα, suggesting that NDMVs not only regulate the expression of inflammatory cytokines but also anti-inflammatory cytokines (such as IL-4) that play regulatory roles in immune function. TNFα up-regulated the expression of G-CSF in our experiments, and this observation is in line with previous reports [41]. However, G-CSF secretion was significantly higher in the TNFα-alone group than the NDMVs + TNFα group. 

In recent years, FLS have been shown to produce a number of cartilage-degrading MMPs [42] as well as a disintegrin and metalloproteinase with thrombospondin motifs (ADAMTS) [43]. ADAMTS-4 and ADAMTS-5 are thought to function as aggrecanase activity responsible for aggrecan degradation in OA [42]. Therefore, it is reasonable to speculate that NDMVs could also induce inactivating or anti-inflammatory functions, suppressed proinflammatory cytokine production and down-regulated cartilage-degrading MMPs and ADAMTS activity. While the down-regulation of inflammatory chemokine and cytokine expression by TNFα-stimulated FLS from OA patients opens new avenues for potential therapeutic approaches, it is clearly important to understand the mechanisms by which the NDMVs regulate this expression. It is possible that their intracellular cargo of proteins and lipids can mediate such effects on gene expression, but it is also possible that they carry microRNA molecules that can regulate gene expression, particularly in TNFα-activated cells [44]. Our hypotheses and preliminary experiments suggest that the microRNA content of NDMVs may have this capacity to regulate gene expression in the target FLS, and we are currently characterizing these molecules and defining their role in this inflammatory process. Recently, Gomez et al. documented that NDMVs could activate NF-κB to elevate inflammatory cytokine levels in endothelial cells from coronary arteries through their uploaded miR-155 [45]. However, the mechanism of NDMVs on target cells could be dissimilar between the proinflammatory effect on endothelial cells and the anti-inflammatory effect on macrophages [17]. Additional studies are underway to further explore the mechanisms of NDMVs in the delivery of microRNAs underlying this phenomenon.

Several inherent limitations of our study merit discussion. First, we established independent FLS cell lines from 15 OA subjects. The cells taken from various donors might behave diversely in the case of immune responses. Second, a mixed pool of FLS was not utilized since heterogeneous cell populations from different donors could be mixed together. However, in the present study, at least five independent experiments with each group were conducted in triplicate in FLS clones from different patients and displayed comparable results. Third, FLS from healthy non-OA subjects were not taken for ethical concerns. Future investigations using commercially available FLS from non-OA patients as controls need to be undertaken to validate our findings. Lastly, proinflammatory cytokine production from FLS treated with TNFα for 6 h was not evaluated. Further studies are warranted to assess the precise effect of NDMVs on cytokine production in TNFα-treated FLS for 6 h.

In conclusion, in contrast to the recognized proinflammatory processes mediated by neutrophils, we disclose, here, that microvesicles derived from TNFα-stimulated human neutrophils can down-regulate cytokine and chemokine expression by FLS isolated from patients with osteoarthritis. This implies that these NDMVs can have an anti-inflammatory effect in certain circumstances, and this observation could shed new insights into the development of new therapeutic strategies to treat this degenerative joint disease.

## 4. Materials and Methods

### 4.1. Primary Culture of Synoviocytes

Synovial tissues were collected from the knees of 15 OA patients diagnosed by the American College of Rheumatology criteria at the time of total knee replacement (Appendix A). Patients with other chronic inflammatory diseases, immunological abnormalities, knee trauma or surgery were excluded in this study. Anti-inflammatory or analgesic drugs were paused for at least 1 week before the surgery. Adipose tissues were excised from phosphate-buffered saline (PBS)-washed synovial tissues prior to mincing of the synovium into 1 mm^2^ pieces which were sequentially digested in Hanks’ Balanced Salt Solution containing 0.25% trypsin (GE Healthcare, Salt Lake City, UT, USA) for 15 min and 1mg/mL collagenase (Sigma-Aldrich, Saint Louis, MI, USA) for 2 h at 37 °C. Double-enzymatic digested tissues were resuspended in Dulbecco’s Modified Eagle Medium (DMEM)/10% fetal bovine serum (FBS) (GE Healthcare, Salt Lake City, UT, USA) and filtered through a 100 µm nylon mesh (Corning, Durham, NC, USA). Dispersed cells were conventionally cultured in DMEM/10%FBS containing 100 U/mL penicillin G, 100 mg/mL streptomycin and 2.5 mg/mL amphotericin B (General Drug House, Bangkok, Thailand) until the fourth or fifth passage (P4 or P5) for experimental use. We established independent primary FLS cell lines from the knee OA patients. Unpooled cells from individual subjects were utilized for subsequent studies.

### 4.2. Flow Cytometry for Synoviocytes Types

The cultured synoviocytes were harvested by 0.25% trypsin for 5 min and adjusted to 1.0 × 10^7^ cells/mL in DMEM/10% FBS. A 100 μL FLS suspension was transferred into 1000 μL PBS/10% FBS and centrifuged at 900× *g* for 3 min. Cell pellets were resuspended in 100 μL PBS/10% FBS. Amounts of 5 µL of FITC-CD90, FITC-CD64, PE-CD55 and APC-CD11b (Biolegend, San Diego, CA, USA) were added, and after the vortex, cell suspensions were incubated at room temperature (RT) in the dark for 30 min. PBS/10% FBS was added at 1000 μL and centrifuged at 900× *g* for 3 min. Cell pellets were resuspended in 500 μL PBS/10% FBS and fixed in 500 μL PBS containing 4% paraformaldehyde at RT for 15 min in the dark. Cell suspensions were pelleted again and finally resuspended in 1000 μL PBS/10% FBS for flow cytometry: BD LSR II (San Jose, CA, USA). When the homogeneity of synoviocytes was >95% CD90+, ≥90% CD55+, <1% CD11b+ and <1% CD64+, the FLS were used for the following experiments (Appendix A).

### 4.3. Neutrophil Isolation

Peripheral blood (50 mL) from healthy volunteers was collected into lithium heparin tubes (Greiner Bio-One, Chonburi, Thailand). Blood was layered over an equal volume of Polymorphprep (Axis-Shield, Oslo, Norway) and centrifuged at 500× *g* for 30 min at 25 °C. After centrifugation, polymorphonuclear leukocytes were collected by pipette, washed with RPMI 1640 medium (Gibco, Grand Island, NY, USA) and centrifuged to remove media and platelets. Contaminating erythrocytes were lysed in red blood cell lysis buffer (Roche, Mannheim, Germany). After centrifugation at 400× *g* for 3 min, pelleted neutrophils were resuspended in RPMI 1640 with 5% vesicle-free serum to give a concentration of 20.0 × 10^6^ cells/mL for subsequent preparation of microvesicles. Isolated neutrophil purity was examined by cytospin and flow cytometry and was typically >97% (Appendix A). CD11b and CD62L expressions on the surface of isolated neutrophils treated with or without TNFα (50 ng/mL) were measured by flow cytometry (Appendix A). Human AB serum (EMD Millipore, Billerica, MA, USA) was ultra-centrifuged at 200,000× *g* for 2 h at 4 °C to remove EVs, and then the supernatant was retained as vesicle-free serum. Neutrophil culture medium was composed of RPMI 1640 and 5% vesicle-free serum, and it was examined by NTA to verify EVs’ disappearance.

### 4.4. NDMVs Release and Isolation

Freshly isolated neutrophils were stimulated by 50 ng/mL recombinant human TNFα (BioLegend, San Diego, CA, USA) for 20 min at 37 °C. The suspension was successively centrifuged at: 4400× *g* for 15 min; 13,000× *g* for 2 min; and 100,000× *g* for 60 min at 4 °C, as previously described by Headland et al. [11]. Pelleted NDMVs (Appendix A) were resuspended in 100 μL particle-free PBS (filtered through a 0.2 µm filter).

### 4.5. Electron Microscopy Analysis of NDMVs

NDMVs were fixed for 3 h in PBS with 2% glutaraldehyde and 2% paraformaldehyde (PFA) prior to analysis by transmission electron microscopy (TEM) and scanning electron microscopy (SEM). For TEM, a copper grid was floated on top of 10 μL of an NDMV suspension for 10 min and washed with one drop of PBS and deionized water. NDMVs attached to the grid were floated on top of 10 μL 2% uranyl acetate and 3% lead citrate to stain for 1 min. The TEM grid was viewed using JEM-1400 Plus TEM operated at 80–100 kV (JEOL, Peabody, MA, USA). For SEM, 5 μL NDMV suspension was placed onto silicon chips and dehydrated with series of solvents for 5 min, each with 100% acetone, 100% ethanol and deionized water. Samples were subjected to critical point drying and mounted on an SEM stub which was sputter coated with 40 nm gold-palladium alloy and observed by JSM-7610F SEM operated at 25–50 kV (JEOL, Peabody, MA, USA).

### 4.6. Nanoparticle Tracking Analysis

NDMVs were diluted 1000-fold in particle-free PBS to a concentration range (10^6^–10^9^ particles/mL), prior to analysis in a NanoSight NS300 (Malvern Panalytical, Malvern, UK). NDMVs’ Brownian motion was recorded in 60 s-long videos (5 times) with the camera level set at 15, under constant flow by a syringe pump (flow rate = 50) at 25 °C. These videos were analyzed automatically using Nanosight NTA 3.1 Software at a detection threshold of 3. For other settings, viscosity was set to “water”, and blur size and max jump distance were “automatic”. Five separate NDMV preparations were analyzed.

### 4.7. MTT Assay

The metabolic activity of NDMV-treated FLS was evaluated by MTT assay (3, (4, 5-dimethylthiazol-2-yl) 2, 5-diphenyl-tetrazolium bromide). A total of 3000 FLS were seeded in a 96-well plate for 24 h and cultured in 60 μL DMEM/10%FBS containing NDMVs at vesicle/cell (V/C) ratios of 0, 10, 30, 50, 100, 200, 300 or 400, or 10 ng/mL TNFα. After incubation for 6, 12, 24 or 48 h, cells were washed with PBS and cultured in 100 μL medium containing 0.5 mg/mL MTT for 4 h at 37 °C. An amount of 100 μL DMSO was added to each well, shaken for 10 min and incubated at 37 °C for 30 min. The formazan absorbance was measured by a Multiskan GO microplate reader (Thermo Fisher Scientific, Waltham, MA, USA) using DMSO-diluted FLS medium (without MTT) as the blank. The viability of FLS without treatment was adjusted to 100% (*n* = 5).

### 4.8. TUNEL Assay

Here, 5.0 × 10^4^ log phase FLS were cultured in 2-well chamber slides (Sigma-Aldrich, Saint Louis, MI, USA) for 24 h of adhesion. FLS were treated with or without NDMVs at V/C ratios of 100 or 300 for 24 h or 48 h in DMEM/10%FBS. Each condition was performed in duplicate. Apoptotic FLS cells were evaluated by terminal deoxynucleotidyl transferase dUTP nick-end labeling (TUNEL) assay using ApopTag Peroxidase In Situ Apoptosis Detection kit (Merck Millipore, Temecula, CA, USA) according to the manufacturer’s protocol. All the stained slides were observed under the light microscope using a computer-supported imaging system. The number of TUNEL-positive cells, defined by cells with dark brown nuclei, was counted by the Aperio ImageScope software (Leica Biosystems Imaging, Inc., Bethesda, MD, USA) and expressed as percentage of positive cells (Appendix A).

### 4.9. Caspase-3 Fluorometric Assay

Caspase-3 activity was assessed using caspase-3 fluorometric assay kit (RayBiotech, Norcross, GA, USA) according to manufacturer’s instructions. Briefly, 2.0 × 10^4^ log phase FLS were seeded in each well of 24-well plates and incubated for 24 h. Adherent FLS were cultured in DMEM/10%FBS with or without NDMVs at V/C ratios of 100 or 300 for 24 h or 48 h. After centrifugation, cell pellets were lysed using lysis buffer on ice and incubated with DEVD-AFC substrate and reaction buffer at 37 °C for 2 h. Caspase-3 activity was analyzed by fluorospectrometry (Varioskan Lux, Thermo Fischer Scientific, Waltham, MA, USA) with appropriate settings (400 nm excitation filter and 505 nm emission filter). The fluorescence intensity of the treated samples was compared with that of control samples to examine the fold-increase in caspase-3 activity. Each condition was performed in triplicate.

### 4.10. Scratch Wound Healing Assay

Briefly, 1.0 × 10^5^ log phase FLS were incubated in 6-well tissue culture plates and incubated for 24 h to form a monolayer. A 200 µL pipette tip was used to scratch a line and remove cells. Each well was washed twice with 1mL DMEM/10%FBS to remove floating cells. An amount of 2 mL of DMEM/10%FBS without or with NDMVs at a V/C ratio of 100 was added to each well, and the plate was incubated for 24 h and 48 h. Cells were washed twice with PBS and fixed with 300 µL 4% PFA for 15 min. The cells were then stained with 300 µL 0.1% crystal violet for 30 min and washed 10 times with PBS. The area of cells between the original scratch lines was evaluated by ImageJ software and % regrowth area was calculated to represent cell proliferation with or without NDMV treatments. The percentage of scratch wound regrowth area was set as a baseline 0% after scratching at 0 h.

### 4.11. Fluorescent Labeling of NDMVs

To aid the quantification of the number of NDMVs taken up by FLS, the phospholipid bilayer membrane of the NDMVs was stained by PKH26 Red Fluorescent Cell Linker Kits for General Cell Membrane Labeling (Sigma-Aldrich, St. Louis, MO, USA). NDMV pellets obtained after ultra-centrifugation were first resuspended gently in 100 μL Diluent C. Equivalent 2×Dye Solution (PKH26 solution/Diluent C ratio of 4 μL:1000 μL) was added to the NDMV mixture and incubated for 5 min at room temperature (RT). An amount of 200 μL EV-free serum was used to stop staining for 1 min at RT. An additional 5 mL RPMI 1640/10% EV-free serum was applied to wash the PKH26-stained NDMVs and dilute residual PKH26. Ultra-centrifugation was used to pellet PKH26-stained NDMVs (Appendix A) at 100,000× *g* for 30 min at 4 °C which were suspended in 100 μL EV-free RPMI 1640. After the size distribution and concentration of PKH26-labeled NDMVs were verified, FLS were treated with NDMVs with an appropriate concentration to measure internalization using flow cytometry and confocal microscopy. 

### 4.12. Flow Cytometry Assay of NDMV Uptake

Briefly, 2.0 × 10^4^ FLS were seeded in a 24-well plate for 24 h and co-cultured with 400 μL DMEM/10%FBS containing NDMVs labeled with PKH26 for time periods from 0.5 to 24 h. Cells were harvested with 0.25% trypsin and fixed in 4% PFA for 15 min at RT. After centrifugation, cells were resuspended in 500 µL PBS, and NDMV uptake was measured by flow cytometry (LSR II, BD Biosciences, Bedford, MA, USA) at 50 mW with a 488 nm blue laser. The fluorescent signals were captured on the PE channel (564–606 nm) and analyzed by FCS Express 4 Reader (De Novo Software, Glendale, CA, USA).

### 4.13. Confocal Microscopy of NDMV Uptake

Before seeding with FLS (as described above), tissue culture-treated glass coverslips (WHB Scientific, Shanghai, China) were placed in the bottom of 24-well plates. After adherence for 24 h at 37 °C, 400 μL DMEM/10% FBS containing labeled NDMVs was added for the indicated time, and then the plate was washed with DMEM/10%FBS. Cells were fixed with 4% PFA for 15 min at RT and then washed in PBS twice. Here, 0.5% Triton X-100 (Thermo Fisher Scientific, Waltham, MA, USA) was used to permeabilize cells for 10 min at RT, and then cells were washed twice with PBS. Non-specific binding was blocked with 5% FBS for 30 min. Flash Phalloidin Green 488 (EMD Millipore, Billerica, MA, USA) at a ratio of 1:100 in PBS was used to stain cellular F-actin for 20 min. Then, 300 nM DAPI solution (EMD Millipore, Billerica, MA, USA) in PBS was used to stain cell nuclei for 5 min. Coverslips were then rinsed with deionized water 3 times and transferred downward on mount slides with a drop of antifade solution (EMD Millipore, Billerica, MA, USA). Slides were viewed by confocal microscopy (LSM800 (Zeiss, Oberkochen, Germany) to capture images.

### 4.14. Multiplex Assay of Inflammatory Cytokines 

Briefly, 1.0 × 10^5^ FLS were cultured in 2 mL DMEM/10%FBS for 24 h and then washed twice in PBS. Adherent cells were treated with NDMVs in DMEM/10% FBS or with and without NDMVs at a V/C ratio of 100 for 6 h and 24 h, ±TNFα at 10 ng/mL. After incubation, cell medium was collected and ultra-centrifugated at 200,000× *g* for 60 min at 4 °C to remove any NDMVs or debris, and supernatants were retained. Cytokine/chemokine levels were measured using the Bio-Plex Pro Human Cytokine 17-plex Assay (Bio-Rad, Hercules, CA, USA), as per the manufacturer’s instructions (detectable range in Appendix A). Briefly, 50 μL diluted magnetic beads was added to each well of an assay plate which was washed twice with Wash Buffer. An amount of 50 µL of culture supernatant was transferred to each well and incubated (with shaking) with beads for 30 min at RT. After washing the plate, 25 μL detection antibody was added and incubated (with shaking) for 30 min at RT. The plate was washed 3 times, and 50 μL diluted SA-PE was added and incubated for 10 min. Lastly, 125 μL assay buffer was added to resuspend the magnetic beads, and the plate was shaken for 30 s. The samples were then measured in a Bio-Plex 200 System (Bio-Rad, Hercules, CA, USA) alongside calibration standards. 

### 4.15. Heatmap and Cluster Analysis

To visualize the expression patterns of the chemokines and cytokines, hierarchical clustering was used to generate a heatmap by the R package “pheatmap”. Cytokine levels in supernatants from 6 different conditions were measured: control, NDMV treatment for 6 h; NDMV treatment for 24 h; NDMV treatment for 6 h plus TNFα (10 ng/mL) 24 h; NDMV treatment for 24 h plus TNFα (10 ng/mL) 24 h; and TNFα (10 ng/mL) for 24 h (*n* = 5). The median values were calculated in these 6 different groups for each cytokine and standardized into a Z-score using a Gaussian normalization equation. Each Z-score was then transformed into a color code.

### 4.16. Statistical Analysis 

Variables were analyzed using R packages (“lsr”, “ggolot”). Quantitative data were presented as mean ± standard error (SE). Non-normally distributed variables were presented as median (quartile 1, quartile 3). The Wilcoxon test was used to determine the differences between two non-normally distributed groups. Statistical differences for more than two groups were assessed using the Kruskal–Wallis non-parametric test followed by multiple comparisons (Conover–Inman test). *p*-Values less than 0.05 were considered as statistically significant.

## Figures and Tables

**Figure 1 ijms-22-07409-f001:**
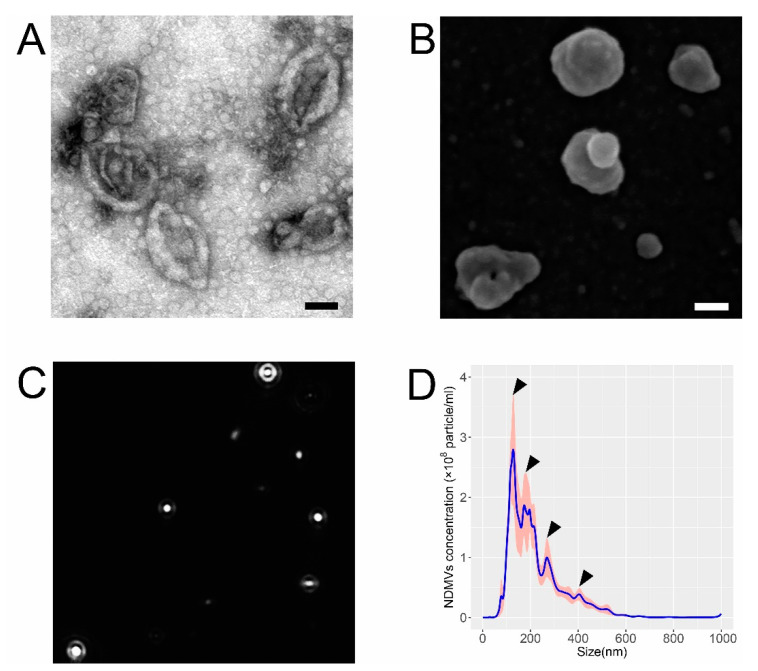
Properties of NDMVs isolated by ultra-centrifugation from TNFα-treated neutrophils. (**A**) NDMVs displayed a cup-shaped appearance in TEM after visualization by negative staining. (**B**) NDMVs appear as irregular spheres with a non-smooth surface in SEM. Scale bar in (**A**,**B**) = 100 nm. (**C**) Screenshot of NTA video that records Brownian motion of NDMVs with highlighted white dots for demonstrating particle size and concentration distribution from one sample in one experiment. (**D**) Size distribution of NDMV size ranging from 100 to 1000 nm after measurements by NTA from 5 separate experiments. The solid line represents the mean concentrations of differently sized NDMVs, with the gray and pink shading showing standard error (SE) of the means (*n* = 5). Arrows indicate each sub-population.

**Figure 2 ijms-22-07409-f002:**
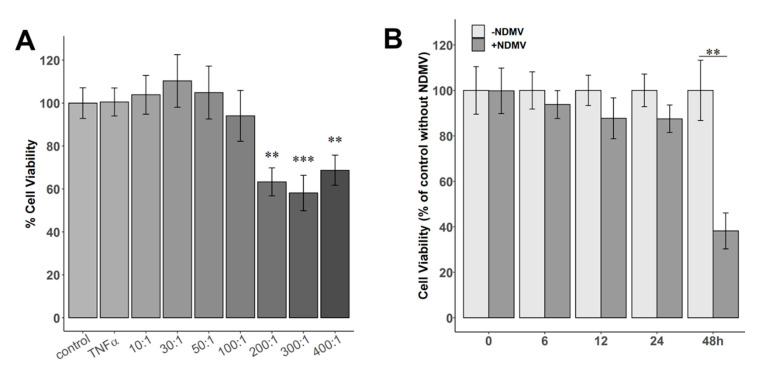
Cell viabilities of FLS incubated with different ratios of NDMVs over time. (**A**) FLS were treated with TNFα alone or NDMVs at V/C ratios of 10, 30, 50, 100, 200, 300 and 400 for 24 h. Control cell viability was designated as 100% for cells incubated in the absence of TNFα and NDMVs after 24 h of incubation for statistical comparison. (**B**) Cell viabilities of FLS incubated in the absence or presence of NDMVs at a ratio of 100 over a 48 h incubation period. Initial cell viability at time 0, 6, 12, 24 or 48 h is defined as 100% for statistical analysis. Significant differences between groups were identified using the Kruskal–Wallis test followed by the Conover–Inman test. Data were analyzed with the Wilcoxon test compared to the non-NDMV-treated group. Values shown are mean (± SE), *n* = 5 separate independent experiments performed in different FLS clones. ** *p* < 0.01, *** *p* < 0.001.

**Figure 3 ijms-22-07409-f003:**
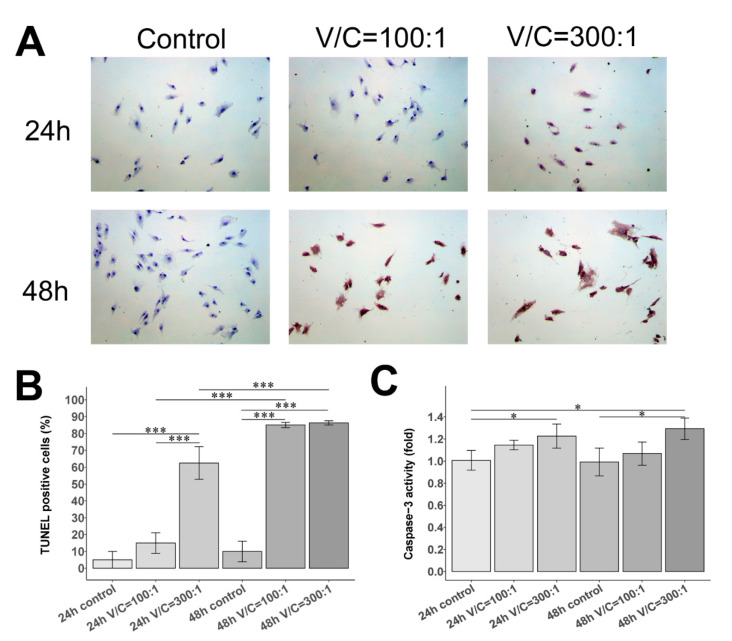
Apoptosis of FLS with NDMV treatment using TUNEL assay and caspase-3 activity assay. (**A**) Representative images of FLS exposed with and without different vesicle/cell (V/C) ratios of 100:1 or 300:1 for 24 h or 48 h by TUNEL assay; ×100 magnification. (**B**) The histogram of TUNEL-positive cells. (**C**) The histogram of caspase-3 activity in control, V/C = 100:1 and V/C = 300:1 groups for 24 h or 48 h. Data are expressed as mean (±SE), *n* = 3 independent experiments performed in different FLS clones. Significant differences between groups were identified using the Kruskal–Wallis test followed by the Conover–Inman test. * *p* < 0.05; *** *p* < 0.001.

**Figure 4 ijms-22-07409-f004:**
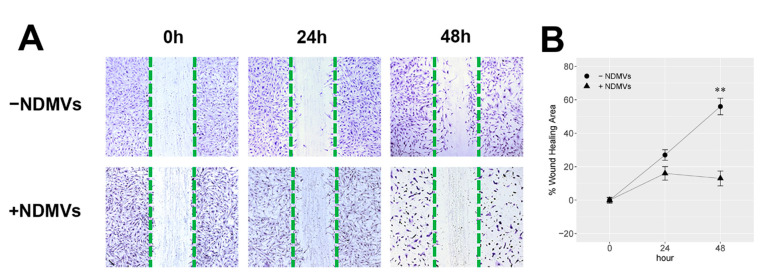
Would healing assay (WHA) for FLS with or without NDMV treatment. (**A**) Images of FLS monolayer scratching with and without NDMVs at 0, 24 and 48 h; (**B**) WHA of FLS incubated in the absence (●) or presence (▲) of NDMVs at a ratio of 100 over 0, 24 and 48 h incubation periods. WHA percentage was quantified by the ImageJ software. Data were analyzed with the Wilcoxon test compared to the non-NDMV-treated group. Values shown are mean (±SE, *n* = 5). ** *p* < 0.01.

**Figure 5 ijms-22-07409-f005:**
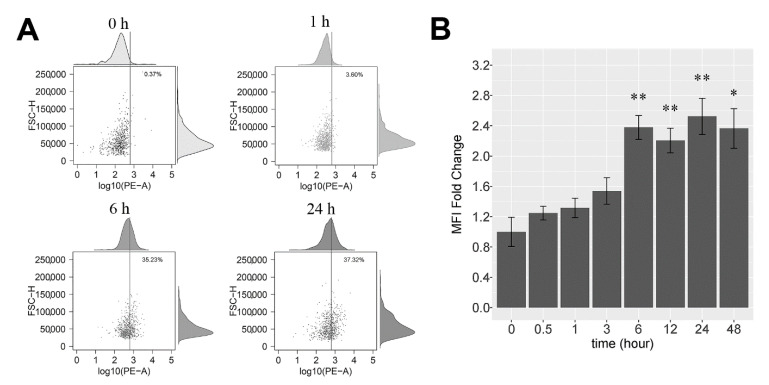
Measurement of NDMV uptake by flow cytometry: (**A**) portrays scatter plot of untreated, control FLS, and 1 h, 6 h and 24 h after addition of NDMVs at a ratio of 100. The vertical line is the gating threshold to identify NDMV-negative and -positive FLS. The upper and right panels are density curves of target fluorescence in FLS and cell size. (**B**) The mean MFI fold change over time of incubation of FLS with labeled NDMVs. Data were normalized to FLS in the absence of NDMVs. Each point represents the mean (±SE) from 5 independent experiments. Significant differences between groups were identified using the Kruskal–Wallis test followed by the Conover–Inman test. * *p* < 0.05, ** *p* < 0.01.

**Figure 6 ijms-22-07409-f006:**
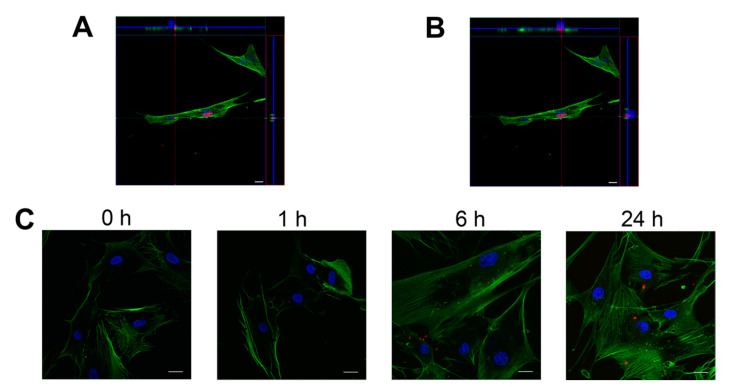
Visualization of NDMVs internalized within FLS by confocal microscopy. NDMVs were incubated with FLS at a ratio of 100. (**A**,**B**) show representative images of three-dimensional analyses after 24 h of incubation. Nuclei were stained by DAPI (blue) and actin by phalloidin (green), whereas NDMVs were labeled by PKH26 (red). Middle images are optical sections; upper and right panels are vertical cross-sections (red and green lines) of NDMVs from confocal Z-stack series demonstrating single and multiple NDMVs in FLS. (**C**) Representative images at 1 h, 6 h and 24 h. Scale bar: 20 μm.

**Figure 7 ijms-22-07409-f007:**
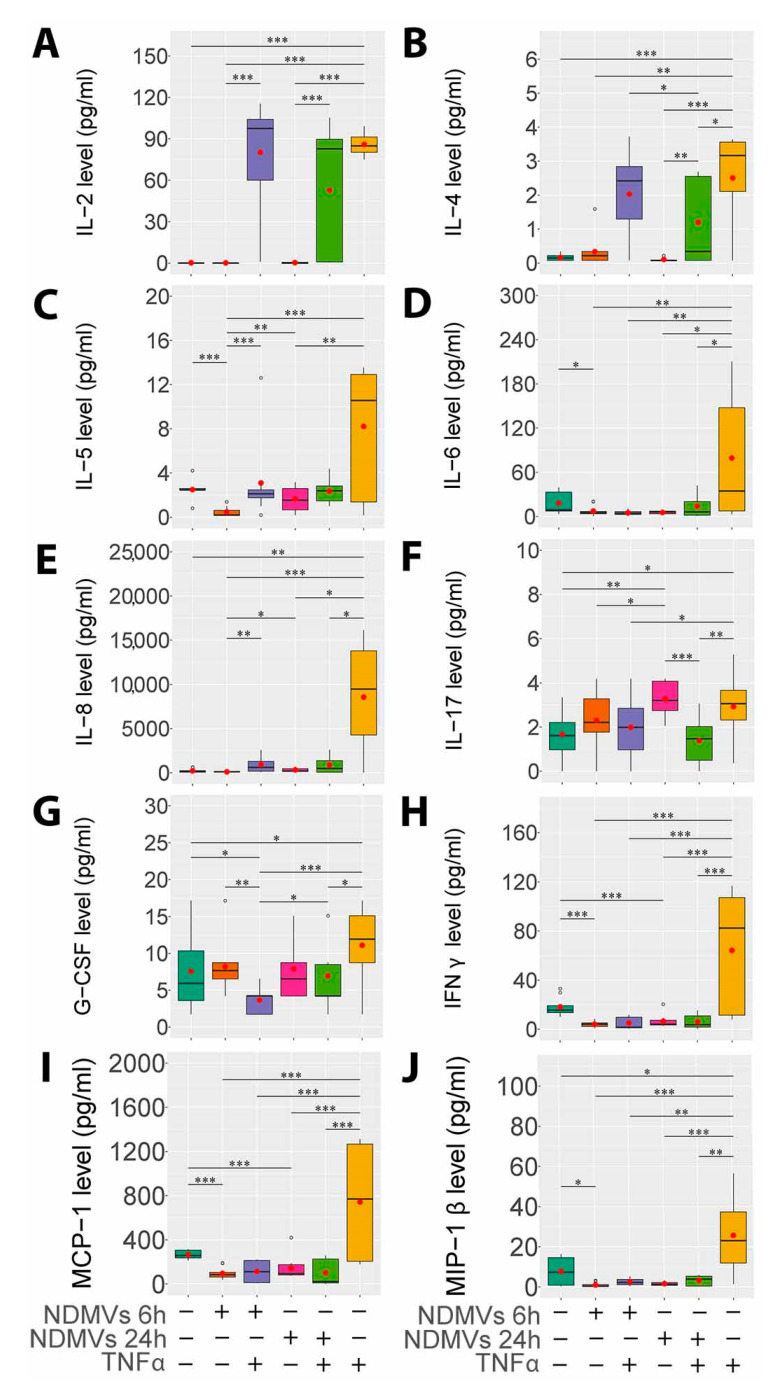
Proinflammatory cytokine and chemokine levels in FLS and NDMV culture supernatants. (**A**, **B**, **C**, **D**, **E**, **F**, **G**, **H**, **I**, **J**) show IL-2, IL-4, IL-5, IL-6, IL-8, IL-17, G-CSF, IFNγ, MCP-1 and MIP-1β levels, respectively, in culture medium after incubation with or without treatment. TNFα also sig-nificantly elevated cytokine levels after incubation are as follows: control; + NDMVs 6 h - TNFα; + NDMVs 24 h - TNFα; + NDMVs 6 h + TNFα; + NDMVs 24 h + TNFα; and + TNFα 24 h. Cytokine levels were analyzed using the Kruskal–Wallis test followed by the Conover–Inman test. Box plots indicate the interquartile ranges (75th to 25th IQR) of the data. The median values are shown as lines within the box, the outlier values are indicated by round circles (◦), and the mean values are indicated by red dots (●). Five independent experiments were performed in different FLS clones. * *p* < 0.05; ** *p* < 0.01; *** *p* < 0.001.

**Figure 8 ijms-22-07409-f008:**
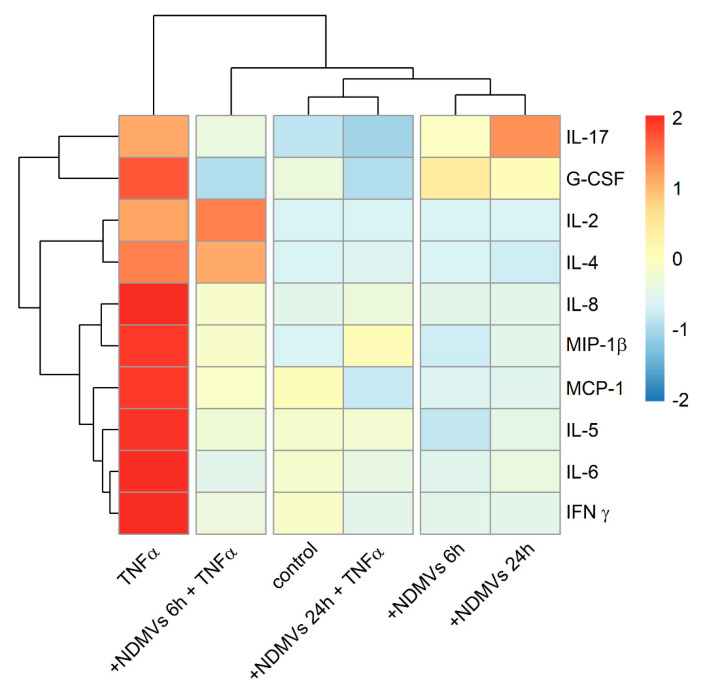
Heatmap of concentrations of proinflammatory cytokines in cultured media of FLS with/without NDMVs and TNFα. Median values were first standardized into Z-score based on each cytokine level in different groups. The hierarchical method was then used for cluster analysis on these standardized data. Color codes represent high (red), intermediate (white) and low (blue) expression.

## Data Availability

The data presented in this study are available on request from the corresponding author. The data are not publicly available due to tissues and cells obtained from patients and healthy person whose privacy rights are protected by laws.

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
