# Peer review of "Internalization of Neutrophil-Derived Microvesicles Modulates TNFα-Stimulated Proinflammatory Cytokine Production in Human Fibroblast-Like Synoviocytes"

_ijms, 2021, doi:10.3390/ijms22147409_

Round 1
Reviewer 1 Report
Title: Neutrophil-derived microvesicles modulate inflammatory cytokine expression in human fibroblast-like synoviocytes
In the current study, authors evaluated effects of Neutrophil-derived microvesicles (NDMVs) on the cytokine production from TNF-treated fibroblast-like synoviocytes (FLS) and on FLS apoptosis exposed to NDMVs. NDMVs affected FLS viability and promoted cell apoptosis when cells were exposed to NDMVs at vesicles/cell (V/C) ratio of over 200 for 48hrs. Therefore, authors treated FLSs with NDMVs at V/C ratio of 100 for 24hrs. Inflammatory cytokine/chemokine production including IL-5, IL-6, IL-8, MCP-1, and MIP-1b by TNFa-treated FLSs were highly impaired when cells were incubated with NDMVs for 24hrs, despite no significant effect on IL-2 and IL-4 production. Although authors focused two functional aspects of NDMVs on cytokine /chemokine production and cell apoptosis in TNF-a-treated FLSs, they do not show experimental evidence to clarify physiological roles of NDMVs on FLS under inflammatory condition in vivo. There are some concerns as below.
Major comments;
- Material and Methods 4.1; Authors prepared FLSs from OA patients, but whether they combined those or established independent cell line for the study were not described in the manuscript. In the latter case, cell differences among patients were raised as serious concerns. Authors need to show comparable data using by FLS clones from different patients. Although authors described numbers of separate experiments in some figure legends, those means the number of attempts in one clone or the number performed in different clones?
- Result 5 “FLS internalize NDMVs in co-culture”; Because Figure 6C presents different cells in 0h, 1h, 6h, 24h, time-dependent internalization of NDMVs by FLS is unable to assess. Authors should track identical cells and present time-lapse image.
- Result 6 “Inflammatory cytokine levels in FLS culture medium”; Box plots with large variations in measured values on each columns indicate that authors combined values from experiments performed in independent FLS clones from different patients. But details are not uncertain. In addition, box plot is not adequate for data presentation in small number of samples.
- Result 6 “Inflammatory cytokine levels in FLS culture medium”; I suppose that authors presented cytokines value from FLSs treated with TNFa for 24hrs in the Figure. 7 rightmost column (NDMVA 6h-, NDMVs24h-, TNFa+), but they did not presented data in FLSs treated with TNFa for 6h. Therefore, the effect of NDMVs on cytokine production in TNFa-treated FLSs for 6h is unable to assess.
- Result 7 “Heatmap and cluster analysis of inflammatory cytokines levels in FLS culture medium”; Because data in Figure 8 is basically same as in Figure.7, Figure 8 is unnecessary for the presentation.
Minor Comments
- Result 1 “Morphological and size characteristics of NDMVs”; Screenshot presented in Figure 1C is not informative. Author should present video to demonstrate Brownian motion of NDMVs in supplemental material.
- Result 1 “Morphological and size characteristics of NDMVs”; Although authors described “NDMVs were identifiable as 4 discrete sub-populations, based on their size (Figure 1D)”, 4 discrete sub-populations are not well identifiable. Arrows indicating each sub-population should need to add in Figure 1D.
- References; The title of Ref.#25 includes hyperlink.
- Figure 2; I do not understand ”(A) FLS were Table 10. 30, 50, 100, 200, 300, and 400 for 24 h.” in figure legend.
- Figure 6; Readers do not understand the difference between (A) and (B) and the importance. Moreover, “Middle images are optical sections; upper and right panels are vertical cross sections (red and green lines) of NDMVs from confocal Z stack series” is not the right legend for (A) and (B).
- Discussion 8th paragraph “In recent years,…”; “ADAMTS-4 and ADAMTS-5 are thought to function as aggrecanase activity responsible for aggrecan degradation in OA42.” “42” just after “degradation in OA” is typographical error.
- Materials and Methods 4.16. ; “Welch' s test” should be corrected to Welch' s t-test.
- Figure S3:” Wilcox” should be typographical error of “Wilcoxon”.
- Authors need to describe statistical method in each figure legend.
Author Response
Reviewer #1:
The manuscript by Zhan D et al., shows Neutrophil-derived microvesicles (NDMVs) modulate inflammatory cytokine expression in fibroblast-like synoviocytes cells from osteoarthritis (OA) patients’ knee. NDMVs were isolated and characterized from tumor necrosis factor-a (TNF-a) stimulated healthy neutrophils. Overall, this study addresses in patients with OA like condition TNF-a induces inflammatory cytokines in synoviocytes, and Parallelly, TNF-a act on neutrophils to produce NDMVs in order to inhibit specific cytokines production from synoviocytes.
Zhan D et al., the finding is interesting to the osteoarthritis translational research community as it includes patient’s data this could also have clinical relevance at some order. However, the author should address the following remarks to improvise the article.
Major points:
- Does Neutrophil-derived microvesicles (NDMVs) contain (NET) neutrophil extracellular traps?
Response: Unfortunately, NET has not been included in our research proposal when it has been submitted and approved with the Institutional Review Board of our institute. Therefore, we could not determine whether NDMVs contain NET.
- Authors mentioned Fibroblast-like synoviocytes (FLS) were isolated from OA patients. If FLS is directly treated with neutrophils (mimicking OA patients’ knees) this will alter NDMVs levels with or without TNF-a.
Response: We cannot conclude that FLS is directly treated with neutrophils this will alter NDMVs levels with or without TNF-a since it was not proposed in our study.
- Page 10 lines 260-264: for Figure 4: reduce the cell size and dark crystal violet staining not clear. Author should provide data of high magnification images and quantification to show cell size and crystal violet staining intensity in Figure 4.
Response: These images are the best we have. We also provide the statistic information in Figure 4.
Minor points:
- Author should merge Figures 7 and 8, as they both contain the same results just displayed in different graph types.
Response: Although figure 7 and 8 both contain the related findings, they displayed the different pattern of the results. Figure 7 shows individual cytokine level alone, whereas figure 8 displays all detectable cytokine expression pattern.
- Figure 6: author should provide high magnification images (or show zoom-in) of PKH26 labeled NDMVs in FLS (Fig. 6a, b) and quantification for Figure 6c.
Response: Although the middle images are identical, upper and right panels are different. We selected different internalized NDMVs to display to confirm vesicle uptake. We addressed this point in the results:
“Confocal microscopy analysis substantiated that the detected fluorescent signal was indeed from the uptake of PKH26-labeled NDMVs by FLS, and not from NDMVs bound to cell surface. Z-stack images were collected to reconstruct three-dimensional images and revealed that single and multiple NDMVs were localized in FLS cytoplasm in equivalent focal planes as the actin cytoskeleton (Figure 6A and 6B). Vertical sections generated by three dimensional analysis of confocal sections revealed that the majority of the fluorescent NDMV signal was indeed observed inside the cells.”
-And in figure 6 legend:
“Middle images are optical sections; upper and right panels are vertical cross sections (red and green lines) of NDMVs from confocal Z stack series showing single and multiple NDMVs in FLS.”

Reviewer 2 Report
The manuscript by Zhan D et al., shows Neutrophil-derived microvesicles (NDMVs) modulate inflammatory cytokine expression in fibroblast-like synoviocytes cells from osteoarthritis (OA) patients’ knee. NDMVs were isolated and characterized from tumor necrosis factor-a (TNF-a) stimulated healthy neutrophils. Overall, this study addresses in patients with OA like condition TNF-a induces inflammatory cytokines in synoviocytes, and Parallelly, TNF-a act on neutrophils to produce NDMVs in order to inhibit specific cytokines production from synoviocytes.
Zhan D et al., the finding is interesting to the osteoarthritis translational research community as it includes patient’s data this could also have clinical relevance at some order. However, the author should address the following remarks to improvise the article.
Major points:
- Does Neutrophil-derived microvesicles (NDMVs) contain (NET) neutrophil extracellular traps?
- Authors mentioned Fibroblast-like synoviocytes (FLS) were isolated from OA patients. If FLS is directly treated with neutrophils (mimicking OA patients’ knees) this will alter NDMVs levels with or without TNF-a.
- Page 10 lines 260-264: for Figure 4: reduce the cell size and dark crystal violet staining not clear. Author should provide data of high magnification images and quantification to show cell size and crystal violet staining intensity in Figure 4.
Minor points:
- Author should merge Figures 7 and 8, as they both contain the same results just displayed in different graph types.
- Figure 6: author should provide high magnification images (or show zoom-in) of PKH26 labeled NDMVs in FLS (Fig. 6a, b) and quantification for Figure 6c.
Author Response
Reviewer #2:
In the present manuscript, the authors investigate the influence of neutrophil-derived microvesicles on the release of inflammatory cytokines. The content of the study is interesting, but some questions remain unanswered, which is why I do not consider the study in its current form suitable for publication in IJMS.
- Synovial fibroblasts from 15 OA patients were used. Are there any differences there compared to the response of SF from non OA patients. The authors should comment on this
Response: -Thank you for your helpful comments and suggestion. Fibroblast-like synoviocytes (FLS) were isolated from synovium which was obtain from OA patients. Synovium is composed of two types of cells: fibroblast-like synoviocytes and macrophage-like synoviocytes. We have to culture in vitro to passage 4 to get high purity of FLS for subsequent studies. The characteristics of FLS from OA patients are virtually similar to FLS from non OA patients. This study used TNF-α for OA stimulation in FLS. Furthermore, it is unethical to obtain FLS from healthy volunteers.
- Was a pool of cells used?
Response: No, it was not. We did not use a pool of cells. Macrophage-like synoviocytes cannot attach on the flask and were washed away during subculture cells. Fibroblast-like synoviocytes (FLS) were left and attached on the flask. We subcultured cells to passage 4 to obtain the high purity of FLS. Meanwhile, the FLS purity of was detected by flow cytometry with fibroblast positive markers-CD90, CD55 and fibroblast negative markers-CD11b, CD64 (Supplementary Figure 1).
- There is no information about the sex and age of the patients
Response: We have already provided the demographic data of the patients in Supplementary Table S1.
- Why is the cell viability limited by the microvesicles. The authors should discuss this finding
Response:
We stated this point in the discussion: “Our studies on the effect of NDMVs on the viability of FLS showed that a greater number of microvesicles predominantly affected the proliferation of the FLS. This process depended on the number of microvesicles added per FLS cell, with a higher vesicle/cell ratio than 100 resulting in a lower cell viabilty. The explanation for this finding could be that NDMVs deliver a range of inflammatory proteins, mRNA, numerous microRNAs and are internalized by FLS and that could influence cell viability in recipient FLS. Additionally, NDMVs may modulate the proliferation of FLS by the generation of reactive oxygen intermediates including the release of MPO and hydrogen peroxide. It should be noted that NDMVs contain a complex mixture of proteins, RNAs, microRNAs, and unknown gene materials that may also contribute to cell viability. Therefore, future studies should address the potential role of these factors. ”
- The standard deviation and not the standard error should be given in the graphs.
Response: We are interested in the precision of the means and in comparing differences between means. Therefore, our data (mean and the standard error) have been given in the graph.
- The statistic is incorrect in most graphs. Since more than two groups are compared here, no T-test or Wilcoxon test should be used.
Response: We used Wilcoxon test because our data in Figure 2B are nonparametric. The nonparametric statistical test (Wilcoxon test) was applied to compare two related, matched groups (between the absence or presence of NDMVs).
- Figure 3 can be integrated into Figure 2 because a similar effect is being studied.
Response: Figure 2 shows cell viability, but figure 3 shows apoptosis. They were conducted with different analyses, therefore they should not be integrated.
- The microscopic images are not representative. Please pick out representative images.
Response: We already changed them.
- Statistic information is missing in fig. 3 -5
Response: We have already provided the statistic information in the legends of Figs. 2-5.
- In Fig. 6, A and B are identical.
Response: No they are not identical. Although the middle images are identical, upper and right panels are different. We selected different internalized NDMVs to display to confirm vesicle uptake. We addressed this point in the results:
“Confocal microscopy analysis substantiated that the detected fluorescent signal was indeed from the uptake of PKH26-labeled NDMVs by FLS, and not from NDMVs bound to cell surface. Z-stack images were collected to reconstruct three-dimensional images and revealed that single and multiple NDMVs were localized in FLS cytoplasm in equivalent focal planes as the actin cytoskeleton (Figure 6A and 6B). Vertical sections generated by three dimensional analysis of confocal sections revealed that the majority of the fluorescent NDMV signal was indeed observed inside the cells.”
-And in figure 6 legend:
“Middle images are optical sections; upper and right panels are vertical cross sections (red and green lines) of NDMVs from confocal Z stack series showing single and multiple NDMVs in FLS.”
- The distribution of the groups and also the statistic comparisons in Fig. 7 do not make any sense and are misleading. Why were 6 h tested here.
Response: According to measurement of NDMVs uptake by flow cytometry (Figure 5A and 5B), by 6 h incubation over 35% of the cells stained positive for NDMVs uptake, while by 24 h incubation, 37% of the FLS stained positive. Mean Fluorescence Intensity (MFI) revealed a 1.25 increase by 0.5 h (compared to controls), rising to the 2.4 folds increase by 6 h, which was largely unaltered by 24 h (2.1-fold) and 48h (2.6-fold) as shown in Figure 5B. The NDMVs internalization by FLS was saturated at 6 h point, which means NDMVs uptake have achieved at maximum and stained at plateau phase. By 6 h incubation, NDMVs could not play optimal effects to inhibit FLS producing inflammatory cytokines. 24 h incubation might be best time to treat FLS for suppress cytokines production.
- The discussion is in large parts a repetition of the results and should be revised.
Response: We have modified and revised the discussion.

Reviewer 3 Report
In the present manuscript, the authors investigate the influence of neutrophil-derived microvesicles on the release of inflammatory cytokines. The content of the study is interesting, but some questions remain unanswered, which is why I do not consider the study in its current form suitable for publication in IJMS.
- Synovial fibroblasts from 15 OA patients were used. Are there any differences there compared to the response of SF from non OA patients. The authors should comment on this
- Was a pool of cells used?
- There is no information about the sex and age of the patients
- Why is the cell viability limited by the microvesicles. The authors should discuss this finding
- the standard deviation and not the standard error should be given in the graphs.
- The statistic is incorrect in most graphs. Since more than two groups are compared here, no T-test or Wilcoxon test should be used.
- Figure 3 can be integrated into Figure 2 because a similar effect is being studied.
- The microscopic images are not representative. Please pick out representative images.
- Statistic information is missing in fig. 3 -5
- In Fig. 6, A and B are identical.
- The distribution of the groups and also the statistic comparisons in Fig. 7 do not make any sense and are misleading. Why were 6 h tested here.
- The discussion is in large parts a repetition of the results and should be revised.
Author Response
Reviewer #3:
In the current study, authors evaluated effects of Neutrophil-derived microvesicles (NDMVs) on the cytokine production from TNF-treated fibroblast-like synoviocytes (FLS) and on FLS apoptosis exposed to NDMVs. NDMVs affected FLS viability and promoted cell apoptosis when cells were exposed to NDMVs at vesicles/cell (V/C) ratio of over 200 for 48hrs. Therefore, authors treated FLSs with NDMVs at V/C ratio of 100 for 24hrs. Inflammatory cytokine/chemokine production including IL-5, IL-6, IL-8, MCP-1, and MIP-1b by TNFa-treated FLSs were highly impaired when cells were incubated with NDMVs for 24hrs, despite no significant effect on IL-2 and IL-4 production. Although authors focused two functional aspects of NDMVs on cytokine /chemokine production and cell apoptosis in TNF-a-treated FLSs, they do not show experimental evidence to clarify physiological roles of NDMVs on FLS under inflammatory condition in vivo. There are some concerns as below.
Major comments:
- Material and Methods 4.1; Authors prepared FLSs from OA patients, but whether they combined those or established independent cell line for the study were not described in the manuscript. In the latter case, cell differences among patients were raised as serious concerns. Authors need to show comparable data using by FLS clones from different patients. Although authors described numbers of separate experiments in some figure legends, those means the number of attempts in one clone or the number performed in different clones?
Response: Thank you very much for valuable suggestion. We addressed this in the methods: “We established primary FLS cell lines from the knee OA patients for subsequent studies.”
- Result 5 “FLS internalize NDMVs in co-culture”; Because Figure 6C presents different cells in 0h, 1h, 6h, 24h, time-dependent internalization of NDMVs by FLS is unable to assess. Authors should track identical cells and present time-lapse image.
Response: Thank you for your comments. We have tried to track identical cells as much as possible.
- Result 6 “Inflammatory cytokine levels in FLS culture medium”; Box plots with large variations in measured values on each columns indicate that authors combined values from experiments performed in independent FLS clones from different patients. But details are not uncertain. In addition, box plot is not adequate for data presentation in small number of samples.
Response: We provided the details in Figure 7: “Cytokine levels were analyzed with Kruskal-Wallis test followed by Conover test: red dots (●) in box plots represent mean values. Five independent experiments were performed in different FLS clones.”
- Result 6 “Inflammatory cytokine levels in FLS culture medium”; I suppose that authors presented cytokines value from FLSs treated with TNFa for 24hrs in the Figure. 7 rightmost column (NDMVA 6h-, NDMVs24h-, TNFa+), but they did not presented data in FLSs treated with TNFa for 6h. Therefore, the effect of NDMVs on cytokine production in TNFa-treated FLSs for 6h is unable to assess.
Response: The effect of NDMVs on cytokine production in TNFa-treated FLSs could be assessed by comparing with non-treated FLSs and NDMVs-treated FLSs for 6h. We provided the details in the results: “The addition of TNFα for 24 h alone significantly stimulated the secretion of IL-2, IL-4, IL-5, IL-6, IL-8, IFNγ, MCP-1, and MIP-1b (P<0.05, Figure 7).”
- Result 7 “Heatmap and cluster analysis of inflammatory cytokines levels in FLS culture medium”; Because data in Figure 8 is basically same as in Figure.7, Figure 8 is unnecessary for the presentation.
Response: Although figure 7 and 8 both contain the related findings, they displayed the different pattern of the results. Figure 7 shows individual cytokine level alone, whereas figure 8 displays all detectable cytokine expression pattern. Therefore, we separate into 2 figures. We provided more details in the results: “Presentation of the levels of the 10 inflammatory cytokines in the form of a color-coded heat map which was generated using hierarchical clustering exhibited a good overview of the profile differences between the groups as depicted in Figure 8.”
Minor comments:
- Result 1 “Morphological and size characteristics of NDMVs”; Screenshot presented in Figure 1C is not informative. Author should present video to demonstrate Brownian motion of NDMVs in supplemental material.
Response: Thank you for valuable suggestion. However, we did not take video in this study.
- Result 1 “Morphological and size characteristics of NDMVs”; Although authors described “NDMVs were identifiable as 4 discrete sub-populations, based on their size (Figure 1D)”, 4 discrete sub-populations are not well identifiable. Arrows indicating each sub-population should need to add in Figure 1D.
Response: We have included arrows indicating each sub-population in Figure 1D.
- References; The title of Ref.#25 includes hyperlink.
Response: We have corrected this.
- Figure 2; I do not understand ”(A) FLS were Table 10. 30, 50, 100, 200, 300, and 400 for 24 h.” in figure legend.
Response: We have corrected this.
- Figure 6; Readers do not understand the difference between (A) and (B) and the importance. Moreover, “Middle images are optical sections; upper and right panels are vertical cross sections (red and green lines) of NDMVs from confocal Z stack series” is not the right legend for (A) and (B).
Response: Although the middle images are identical, upper and right panels are different. We selected different internalized NDMVs to display to confirm vesicle uptake. We addressed this point in the results:
“Confocal microscopy analysis substantiated that the detected fluorescent signal was indeed from the uptake of PKH26-labeled NDMVs by FLS, and not from NDMVs bound to cell surface. Z-stack images were collected to reconstruct three-dimensional images and revealed that single and multiple NDMVs were localized in FLS cytoplasm in equivalent focal planes as the actin cytoskeleton (Figure 6A and 6B). Vertical sections generated by three dimensional analysis of confocal sections revealed that the majority of the fluorescent NDMV signal was indeed observed inside the cells.”
-And in figure 6 legend:
“Middle images are optical sections; upper and right panels are vertical cross sections (red and green lines) of NDMVs from confocal Z stack series showing single and multiple NDMVs in FLS.”
- Discussion 8th paragraph “In recent years,…”; “ADAMTS-4 and ADAMTS-5 are thought to function as aggrecanase activity responsible for aggrecan degradation in OA42.” “42” just after “degradation in OA” is typographical error.
Response: We have corrected this.
- Materials and Methods 4.16. ; “Welch' s test” should be corrected to Welch' s t-test.
Response: We have corrected this.
- Figure S3:” Wilcox” should be typographical error of “Wilcoxon”.
Response: We have corrected this.
- Authors need to describe statistical method in each figure legend.
Response: We have already provided the statistic information in the legends of Figs.

Round 2
Reviewer 1 Report
Authors assessed apoptosis, internalization of neutrophil-derived microvesicles, therefore, the manuscript title "Neutrophil-derived microvesicles modulate inflammatory cytokine expression in human fibroblast-like synoviocytes" partially expresses the study.
Author Response
Authors assessed apoptosis, internalization of Neutrophil-derived microvesicles, therefore, the manuscript title “Neutrophil-derived microvesicles modulate inflammatory cytokine expression in human fibroblast-like synoviocytes” partially expresses the study.
Response: Thank you very much for your suggestion. We modified the manuscript title as follows:
“Internalization of neutrophil-derived microvesicles modulates TNFα-stimulated proinflammatory cytokine production in human fibroblast-like synoviocytes”
Reviewer 3 Report
Unfortunately, my comments were not addressed (see below). Please complete and resubmit.
In the present manuscript, the authors investigate the influence of neutrophil-derived microvesicles on the release of inflammatory cytokines. The content of the study is interesting, but some questions remain unanswered, which is why I do not consider the study in its current form suitable for publication in IJMS.
- Synovial fibroblasts from 15 OA patients were used. Are there any differences there compared to the response of SF from non OA patients. The authors should comment on this
- Was a pool of cells used?
- There is no information about the sex and age of the patients
- Why is the cell viability limited by the microvesicles. The authors should discuss this finding
- the standard deviation and not the standard error should be given in the graphs.
- The statistic is incorrect in most graphs. Since more than two groups are compared here, no T-test or Wilcoxon test should be used.
- Figure 3 can be integrated into Figure 2 because a similar effect is being studied.
- The microscopic images are not representative. Please pick out representative images.
- Statistic information is missing in fig. 3 -5
- In Fig. 6, A and B are identical.
- The distribution of the groups and also the statistic comparisons in Fig. 7 do not make any sense and are misleading. Why were 6 h tested here.
- The discussion is in large parts a repetition of the results and should be revised.
Author Response
In the present manuscript, the authors investigate the influence of neutrophil-derived microvesicles on the release of inflammatory cytokines. The content of the study is interesting, but some questions remain unanswered, which is why I do not consider the study in its current form suitable for publication in IJMS.
- Synovial fibroblasts from 15 OA patients were used. Are there any differences there compared to the response of SF from non OA patients. The authors should comment on this
Response: Thank you very much for your helpful comments and kind suggestion. Fibroblast-like synoviocytes (FLS) were isolated from synovium which was obtain from OA patients. Synovium is composed of two types of cells: fibroblast-like synoviocytes and macrophage-like synoviocytes. We have to culture in vitro to passage 4 to get high purity of FLS for subsequent studies. The characteristics of FLS from OA patients are virtually similar to FLS from non OA patients. This study used TNF-α for OA stimulation in FLS. Furthermore, it is unethical to obtain FLS from healthy volunteers.
- Was a pool of cells used?
Response: No, it was not. We did not use a pool of cells. Macrophage-like synoviocytes cannot attach on the flask and were washed away during subculture cells. Fibroblast-like synoviocytes (FLS) were left and attached on the flask. We subcultured cells to passage 4 to obtain the high purity of FLS. Meanwhile, the FLS purity of was detected by flow cytometry with fibroblast positive markers-CD90, CD55 and fibroblast negative markers-CD11b, CD64 (Supplementary Figure 1).
- There is no information about the sex and age of the patients
Response: We have already provided the demographic data of the patients in Supplementary Table S1.
- Why is the cell viability limited by the microvesicles. The authors should discuss this finding
Response:
We stated this point in the discussion: “Our studies on the effect of NDMVs on the viability of FLS showed that a greater number of microvesicles predominantly affected the proliferation of the FLS. This process depended on the number of microvesicles added per FLS cell, with a higher vesicle/cell ratio than 100 resulting in a lower cell viabilty. The explanation for this finding could be that NDMVs deliver a range of inflammatory proteins, mRNA, numerous microRNAs and are internalized by FLS and that could influence cell viability in recipient FLS. Additionally, NDMVs may modulate the proliferation of FLS by the generation of reactive oxygen intermediates including the release of MPO and hydrogen peroxide. It should be noted that NDMVs contain a complex mixture of proteins, RNAs, microRNAs, and unknown gene materials that may interfere with the deregulated FLS survival. Therefore, future studies should address the potential role of these factors.”
- The standard deviation and not the standard error should be given in the graphs.
Response: We are interested in the precision of the means and in comparing differences between means. Therefore, our data (mean and the standard error) have been given in the graph.
- The statistic is incorrect in most graphs. Since more than two groups are compared here, no T-test or Wilcoxon test should be used.
Response: Thank you very much for your suggestion. We already checked and corrected as shown in the statistical analysis and figure legends.
“Non-normally distributed variables were presented as median (quartile 1, quartile 3). Wilcoxon test was used to determine the differences between two non-normally distributed groups. Statistical differences for more than two groups were assessed using Kruskal-Wallis non-parametric test followed by multiple comparisons (Conover-Inman) test.”
- Figure 3 can be integrated into Figure 2 because a similar effect is being studied.
Response: Figure 2 shows cell viability, but figure 3 shows apoptosis. They were conducted with different analyses, therefore they should not be integrated.
- The microscopic images are not representative. Please pick out representative images.
Response: We already changed them.
- Statistic information is missing in fig. 3 -5
Response: We have already provided the statistic information in the legends of Figs. 2-5.
- In Fig. 6, A and B are identical.
Response: No they are not identical. Although the middle images are identical, the upper and right panels are different. We selected different internalized NDMVs to display in order to confirm vesicle uptake. We addressed this point in the results:
“Confocal microscopy analysis substantiated that the detected fluorescent signal was indeed from the uptake of PKH26-labeled NDMVs by FLS, and not from NDMVs bound to cell surface. Z-stack images were collected to reconstruct three-dimensional images and revealed that single and multiple NDMVs were localized in FLS cytoplasm in equivalent focal planes as the actin cytoskeleton (Figure 6A and 6B). Vertical sections generated by three dimensional analysis of confocal sections affirmed that the majority of the fluorescent NDMV signal was indeed observed inside the cells.”
-And in figure 6 legend:
“Middle images are optical sections; upper and right panels are vertical cross sections (red and green lines) of NDMVs from confocal Z stack series demonstrating single and multiple NDMVs in FLS.”
- The distribution of the groups and also the statistic comparisons in Fig. 7 do not make any sense and are misleading. Why were 6 h tested here.
Response: According to measurement of NDMVs uptake by flow cytometry (Figure 5A and 5B), by 6 h incubation over 35% of the cells stained positive for NDMVs uptake, while by 24 h incubation, 37% of the FLS stained positive. Mean Fluorescence Intensity (MFI) revealed a 1.25 increase by 0.5 h (compared to controls), rising to the 2.4 folds increase by 6 h, which was largely unaltered by 24 h (2.1-fold) and 48h (2.6-fold) as shown in Figure 5B. The NDMVs internalization by FLS was saturated at 6 h point, which means NDMVs uptake have achieved at maximum and stained at plateau phase. By 6 h incubation, NDMVs could not play optimal effects to inhibit FLS producing inflammatory cytokines. 24 h incubation might be best time to treat FLS for suppress cytokines production.
- The discussion is in large parts a repetition of the results and should be revised.
Response: We have modified and revised the discussion. We also included the limitation of this study as shown in the discussion:
“Several inherent limitations of our study merit discussion. First, we established independent FLS cell lines from 15 OA subjects. The cells taken from various donors might behave diversely in case of immune responses. Second, a mixed pool of FLS was not utilized since heterogeneous cell populations from different donors could be mixed together. However, in the present study, at least five independent experiments with each group were conducted in triplicate in FLS clones from different patients and displayed comparable results. Third, FLS from healthy non-OA subjects were not taken for ethical concerns. Therefore, the response of FLS from non-OA patients may not be determined. Lastly, proinflammatory cytokine production from FLS treated with TNFα for 6 h was not evaluated. Further studies are warranted to assess the precise effect of NDMVs on cytokine production in TNFα-treated FLS for 6h”

Round 3
Reviewer 3 Report
The authors have answered most of my questions. Only the question about the cells used remains open. FLS from 15 patients were isolated. Were these cell lines thrown together(pooled), or considered in isolation. FYI, there are commercially available FLS from non-OA patients for control.
Author Response
Reviewer #3:
The authors have answered most of my questions. Only the question about the cells used remains open. FLS from 15 patients were isolated. Were these cell lines thrown together(pooled), or considered in isolation. FYI, there are commercially available FLS from non-OA patients for control.
Response:
Thank you very much for valuable perspective. We agree with the reviewer comments and include additional information in the methods. We also address this point in the discussion:
“We established independent primary FLS cell lines from the knee OA patients. Unpooled cells from individual subjects were utilized for subsequent studies.”
“Second, a mixed pool of FLS was not utilized since heterogeneous cell populations from different donors could be mixed together. However, in the present study, at least five independent experiments with each group were conducted in triplicate in FLS clones from different patients and displayed comparable results. Third, FLS from healthy non-OA subjects were not taken for ethical concerns. Future investigations using commercially available FLS from non-OA patients as controls need to be undertaken to validate our findings.”
